# Decision-Based Adversarial Attacks: Reliable Attacks Against Black-Box Machine Learning Models

**Wieland Brendel**[∗]**, Jonas Rauber**[∗] **& Matthias Bethge**
Werner Reichardt Centre for Integrative Neuroscience,
Eberhard Karls University Tübingen, Germany
{wieland,jonas,matthias}@bethgelab.org

## Abstract

Many machine learning algorithms are vulnerable to almost imperceptible perturbations of their inputs. So far it was unclear how much risk adversarial perturbations carry for the safety of real-world machine learning applications because most methods used to generate such perturbations rely either on detailed model information (*gradient-based attacks*) or on confidence scores such as class probabilities (*score-based attacks*), neither of which are available in most real-world scenarios. In many such cases one currently needs to retreat to *transfer-based attacks* which rely on cumbersome substitute models, need access to the training data and can be defended against. Here we emphasise the importance of attacks which solely rely on the final model decision. Such *decision-based attacks* are (1) applicable to real-world black-box models such as autonomous cars, (2) need less knowledge and are easier to apply than transfer-based attacks and (3) are more robust to simple defences than gradient- or score-based attacks. Previous attacks in this category were limited to simple models or simple datasets. Here we introduce the Boundary Attack, a decision-based attack that starts from a large adversarial perturbation and then seeks to reduce the perturbation while staying adversarial. The attack is conceptually simple, requires close to no hyperparameter tuning, does not rely on substitute models and is competitive with the best gradient-based attacks in standard computer vision tasks like ImageNet. We apply the attack on two black-box algorithms from Clarifai.com. The Boundary Attack in particular and the class of decision-based attacks in general open new avenues to study the robustness of machine learning models and raise new questions regarding the safety of deployed machine learning systems. An implementation of the attack is available as part of Foolbox (https://github.com/bethgelab/foolbox).

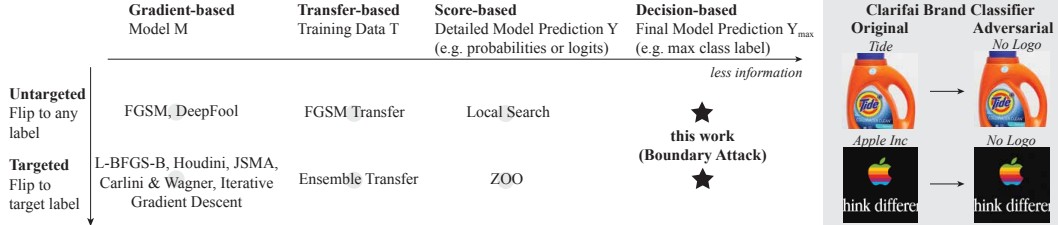

Figure 1: (Left) Taxonomy of adversarial attack methods. The Boundary Attack is applicable to real-world ML algorithms because it only needs access to the final decision of a model (e.g. class-label or transcribed sentence) and does not rely on model information like the gradient or the confidence scores. (Right) Application to the Clarifai Brand Recognition Model.

---

[∗]Equal contribution.

## 1 INTRODUCTION

Many high-performance machine learning algorithms used in computer vision, speech recognition and other areas are susceptible to minimal changes of their inputs (Szegedy et al., 2013). As a concrete example, a modern deep neural network like VGG-19 trained on object recognition might perfectly recognize the main object in an image as a *tiger cat*, but if the pixel values are only slightly perturbed in a specific way then the prediction of the very same network is drastically altered (e.g. to *bus*). These so-called adversarial perturbations are ubiquitous in many machine learning models and are often imperceptible to humans. Algorithms that seek to find such adversarial perturbations are generally denoted as *adversarial attacks*.

Adversarial perturbations have drawn interest from two different sides. On the one side, they are worrisome for the integrity and security of deployed machine learning algorithms such as autonomous cars or face recognition systems. Minimal perturbations on street signs (e.g. turning a stop-sign into a 200 km/h speed limit) or street lights (e.g. turning a red into a green light) can have severe consequences. On the other hand, adversarial perturbations provide an exciting spotlight on the gap between the sensory information processing in humans and machines and thus provide guidance towards more robust, human-like architectures.

Adversarial attacks can be roughly divided into three categories: *gradient-based*, *score-based* and *transfer-based* attacks (cp. Figure 1). Gradient-based and score-based attacks are often denoted as white-box and oracle attacks respectively, but we try to be as explicit as possible as to what information is being used in each category[1]. A severe problem affecting attacks in all of these categories is that they are surprisingly straight-forward to defend against:

- **Gradient-based attacks.** Most existing attacks rely on detailed model information including the gradient of the loss w.r.t. the input. Examples are the Fast-Gradient Sign Method (FGSM), the Basic Iterative Method (BIM) (Kurakin et al., 2016), DeepFool (Moosavi-Dezfooli et al., 2015), the Jacobian-based Saliency Map Attack (JSMA) (Papernot et al., 2015), Houdini (Cisse et al., 2017) and the Carlini & Wagner attack (Carlini & Wagner, 2016a).

  *Defence:* A simple way to defend against gradient-based attacks is to mask the gradients, for example by adding non-differentiable elements either implicitly through means like defensive distillation (Papernot et al., 2016) or saturated non-linearities (Nayebi & Ganguli, 2017), or explicitly through means like non-differentiable classifiers (Lu et al., 2017).

- **Score-based attacks.** A few attacks are more agnostic and only rely on the predicted scores (e.g. class probabilities or logits) of the model. On a conceptual level these attacks use the predictions to numerically estimate the gradient. This includes black-box variants of JSMA (Narodytska & Kasiviswanathan, 2016) and of the Carlini & Wagner attack (Chen et al., 2017) as well as generator networks that predict adversarials (Hayes & Danezis, 2017).

  *Defence:* It is straight-forward to severely impede the numerical gradient estimate by adding stochastic elements like dropout into the model. Also, many robust training methods introduce a sharp-edged plateau around samples (Tramer et al., 2017) which not only masks gradients themselves but also their numerical estimate.

- **Transfer-based attacks.** Transfer-based attacks do not rely on model information but need information about the training data. This data is used to train a fully observable substitute model from which adversarial perturbations can be synthesized (Papernot et al., 2017a). They rely on the empirical observation that adversarial examples often transfer between models. If adversarial examples are created on an ensemble of substitute models the success rate on the attacked model can reach up to 100% in certain scenarios (Liu et al., 2016).

  *Defence:* A recent defence method against transfer attacks (Tramer et al., 2017), which is based on robust training on a dataset augmented by adversarial examples from an ensemble of substitute models, has proven highly successful against basically all attacks in the 2017 Kaggle Competition on Adversarial Attacks[2].

---

[1]For example, the term *oracle* does not convey what information is used by attacks in this category.
[2]https://www.kaggle.com/c/nips-2017-defense-against-adversarial-attack

The fact that many attacks can be easily averted makes it often extremely difficult to assess whether a model is truly robust or whether the attacks are just too weak, which has lead to premature claims of robustness for DNNs (Carlini & Wagner, 2016b; Brendel & Bethge, 2017).

This motivates us to focus on a category of adversarial attacks that has so far received fairly little attention:

- **Decision-based attacks.** Direct attacks that solely rely on the final decision of the model (such as the top-1 class label or the transcribed sentence).

The delineation of this category is justified for the following reasons: First, compared to score-based attacks decision-based attacks are much more relevant in real-world machine learning applications where confidence scores or logits are rarely accessible. At the same time decision-based attacks have the potential to be much more robust to standard defences like gradient masking, intrinsic stochasticity or robust training than attacks from the other categories. Finally, compared to transfer-based attacks they need much less information about the model (neither architecture nor training data) and are much simpler to apply.

There currently exists no effective decision-based attack that scales to natural datasets such as ImageNet and is applicable to deep neural networks (DNNs). The most relevant prior work is a variant of transfer attacks in which the training set needed to learn the substitute model is replaced by a synthetic dataset (Papernot et al., 2017b). This synthetic dataset is generated by the adversary alongside the training of the substitute; the labels for each synthetic sample are drawn from the black-box model. While this approach works well on datasets for which the intra-class variability is low (such as MNIST) it has yet to be shown that it scales to more complex natural datasets such as CIFAR or ImageNet. Other decision-based attacks are specific to linear or convex-inducing classifiers (Dalvi et al., 2004; Lowd & Meek, 2005; Nelson et al., 2012) and are not applicable to other machine learning models. The work by (Biggio et al., 2013) basically stands between transfer attacks and decision-based attacks in that the substitute model is trained on a dataset for which the labels have been observed from the black-box model. This attack still requires knowledge about the data distribution on which the black-box models was trained on and so we don't consider it a pure decision-based attack. Finally, some naive attacks such as a line-search along a random direction away from the original sample can qualify as decision-based attacks but they induce large and very visible perturbations that are orders of magnitude larger than typical gradient-based, score-based or transfer-based attacks.

Throughout the paper we focus on the threat scenario in which the adversary aims to change the decision of a model (either targeted or untargeted) for a particular input sample by inducing a minimal perturbation to the sample. The adversary can observe the final decision of the model for arbitrary inputs and it knows at least one perturbation, however large, for which the perturbed sample is adversarial.

The contributions of this paper are as follows:

- We emphasise decision-based attacks as an important category of adversarial attacks that are highly relevant for real-world applications and important to gauge model robustness.

- We introduce the first effective decision-based attack that scales to complex machine learning models and natural datasets. The Boundary Attack is (1) conceptually surprisingly simple, (2) extremely flexible, (3) requires little hyperparameter tuning and (4) is competitive with the best gradient-based attacks in both targeted and untargeted computer vision scenarios.

- We show that the Boundary Attack is able to break previously suggested defence mechanisms like defensive distillation.

- We demonstrate the practical applicability of the Boundary Attack on two black-box machine learning models for brand and celebrity recognition available on Clarifai.com.

## 1.1 NOTATION

Throughout the paper we use the following notation: $\boldsymbol{o}$ refers to the original input (e.g. an image), $\boldsymbol{y} = F(\boldsymbol{o})$ refers to the full prediction of the model $F(\cdot)$ (e.g. logits or probabilities), $y_{max}$ is the

predicted label (e.g. class-label). Similarly, $\tilde{\boldsymbol{o}}$ refers to the adversarially perturbed image, $\tilde{\boldsymbol{o}}^k$ refers to the perturbed image at the $k$-th step of an attack algorithm. Vectors are denoted in bold.

## 2 BOUNDARY ATTACK

The basic intuition behind the boundary attack algorithm is depicted in Figure 2: the algorithm is *initialized* from a point that is already adversarial and then performs a random walk along the boundary between the adversarial and the non-adversarial region such that (1) it stays in the adversarial region and (2) the distance towards the target image is reduced. In other words we perform rejection sampling with a suitable *proposal distribution* $\mathcal{P}$ to find progressively smaller adversarial perturbations according to a given *adversarial criterion* $c(.)$. The basic logic of the algorithm is described in Algorithm 1, each individual building block is detailed in the next subsections.

---

**Data:** original image $\mathbf{o}$, adversarial criterion $c(.)$, decision of model $d(.)$
**Result:** adversarial example $\tilde{\boldsymbol{o}}$ such that the distance $d(\boldsymbol{o}, \tilde{\boldsymbol{o}}) = \|\boldsymbol{o} - \tilde{\boldsymbol{o}}\|_2^2$ is minimized
initialization: $k = 0$, $\tilde{\boldsymbol{o}}^0 \sim \mathcal{U}(0, 1)$ s.t. $\tilde{\boldsymbol{o}}^0$ is adversarial;
**while** $k <$ *maximum number of steps* **do**
    draw random perturbation from proposal distribution $\boldsymbol{\eta}_k \sim \mathcal{P}(\tilde{\boldsymbol{o}}^{k-1})$;
    **if** $\tilde{\boldsymbol{o}}^{k-1} + \boldsymbol{\eta}_k$ *is adversarial* **then**
        set $\tilde{\boldsymbol{o}}^k = \tilde{\boldsymbol{o}}^{k-1} + \boldsymbol{\eta}_k$;
    **else**
        set $\tilde{\boldsymbol{o}}^k = \tilde{\boldsymbol{o}}^{k-1}$;
    **end**
    $k = k + 1$
**end**

**Algorithm 1:** Minimal version of the Boundary Attack.

---

### 2.1 INITIALISATION

The Boundary Attack needs to be initialized with a sample that is already adversarial[3]. In an untargeted scenario we simply sample from a maximum entropy distribution given the valid domain of the input. In the computer vision applications below, where the input is constrained to a range of $[0, 255]$ per pixel, we sample each pixel in the initial image $\tilde{\mathbf{o}}^0$ from a uniform distribution $\mathcal{U}(0, 255)$. We reject samples that are not adversarial. In a targeted scenario we start from any sample that is classified by the model as being from the target class.

### 2.2 PROPOSAL DISTRIBUTION

The efficiency of the algorithm crucially depends on the proposal distribution $\mathcal{P}$, i.e. which random directions are explored in each step of the algorithm. The optimal proposal distribution will generally depend on the domain and / or model to be attacked, but for all vision-related problems tested here a very simple proposal distribution worked surprisingly well. The basic idea behind this proposal distribution is as follows: in the $k$-th step we want to draw perturbations $\boldsymbol{\eta}^k$ from a maximum entropy distribution subject to the following constraints:

1. The perturbed sample lies within the input domain,
$$\tilde{o}_i^{k-1} + \eta_i^k \in [0, 255]. \tag{1}$$

2. The perturbation has a relative size of $\delta$,
$$\left\| \boldsymbol{\eta}^k \right\|_2 = \delta \cdot d(\mathbf{o}, \tilde{\mathbf{o}}^{k-1}). \tag{2}$$

3. The perturbation reduces the distance of the perturbed image towards the original input by a relative amount $\epsilon$,
$$d(\mathbf{o}, \tilde{\mathbf{o}}^{k-1}) - d(\mathbf{o}, \tilde{\mathbf{o}}^{k-1} + \boldsymbol{\eta}^k) = \epsilon \cdot d(\mathbf{o}, \tilde{\mathbf{o}}^{k-1}). \tag{3}$$

---

[3]Note that here *adversarial* does not mean that the decision of the model is wrong—it might make perfect sense to humans—but that the perturbation fulfills the adversarial criterion (e.g. changes the model decision).

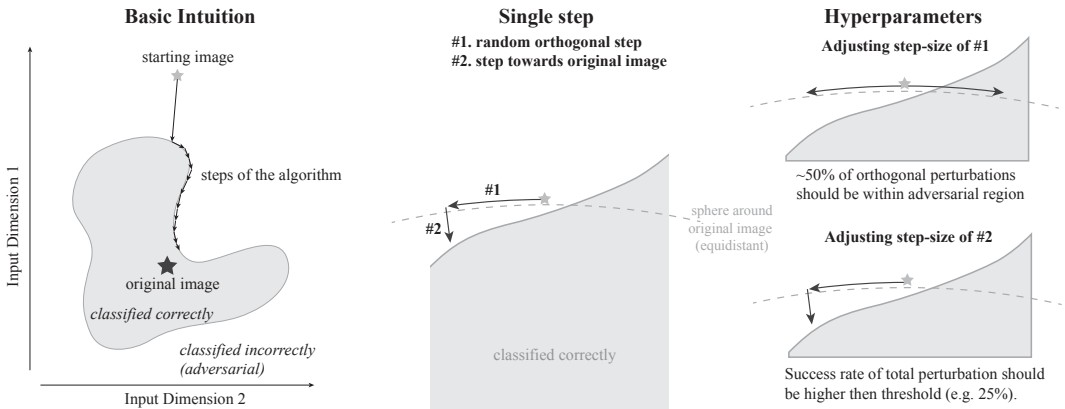

Figure 2: (Left) In essence the Boundary Attack performs rejection sampling along the boundary between adversarial and non-adversarial images. (Center) In each step we draw a new random direction by (#1) drawing from an iid Gaussian and projecting on a sphere, and by (#2) making a small move towards the target image. (Right) The two step-sizes (orthogonal and towards the original input) are dynamically adjusted according to the local geometry of the boundary.

In practice it is difficult to sample from this distribution, and so we resort to a simpler heuristic: first, we sample from an iid Gaussian distribution $\eta_i^k \sim \mathcal{N}(0, 1)$ and then rescale and clip the sample such that (1) and (2) hold. In a second step we project $\eta^k$ onto a sphere around the original image $o$ such that $d(o, \tilde{o}^{k-1} + \eta^k) = d(o, \tilde{o}^{k-1})$ and (1) hold. We denote this as the *orthogonal perturbation* and use it later for hyperparameter tuning. In the last step we make a small movement towards the original image such that (1) and (3) hold. For high-dimensional inputs and small $\delta, \epsilon$ the constraint (2) will also hold approximately.

## 2.3 ADVERSARIAL CRITERION

A typical criterion by which an input is classified as *adversarial* is misclassification, i.e. whether the model assigns the perturbed input to some class different from the class label of the original input. Another common choice is targeted misclassification for which the perturbed input has to be classified in a given target class. Other choices include top-k misclassification (the top-k classes predicted for the perturbed input do not contain the original class label) or thresholds on certain confidence scores. Outside of computer vision many other choices exist such as criteria on the word-error rates. In comparison to most other attacks, the Boundary Attack is extremely flexible with regards to the adversarial criterion. It basically allows any criterion (including non-differentiable ones) as long as for that criterion an initial adversarial can be found (which is trivial in most cases).

## 2.4 HYPERPARAMETER ADJUSTMENT

The Boundary Attack has only two relevant parameters: the length of the total perturbation $\delta$ and the length of the step $\epsilon$ towards the original input (see Fig. 2). We adjust both parameters dynamically according to the local geometry of the boundary. The adjustment is inspired by Trust Region methods. In essence, we first test whether the orthogonal perturbation is still adversarial. If this is true, then we make a small movement towards the target and test again. The orthogonal step tests whether the step-size is small enough so that we can treat the decision boundary between the adversarial and the non-adversarial region as being approximately linear. If this is the case, then we expect around 50% of the orthogonal perturbations to still be adversarial. If this ratio is much lower, we reduce the step-size $\delta$, if it is close to 50% or higher we increase it. If the orthogonal perturbation is still adversarial we add a small step towards the original input. The maximum size of this step depends on the angle of the decision boundary in the local neighbourhood (see also Figure 2). If the success rate is too small we decrease $\epsilon$, if it is too large we increase it. Typically, the closer we get to the original image, the flatter the decision boundary becomes and the smaller $\epsilon$ has to be to still make progress. The attack is converged whenever $\epsilon$ converges to zero.

## 3  COMPARISON WITH OTHER ATTACKS

We quantify the performance of the Boundary Attack on three different standard datasets: MNIST (LeCun et al., 1998), CIFAR-10 (Krizhevsky & Hinton, 2009) and ImageNet-1000 (Deng et al., 2009). To make the comparison with previous results as easy and transparent as possible, we here use the same MNIST and CIFAR networks as Carlini & Wagner (2016a)[4]. In a nutshell, both the MNIST and CIFAR model feature nine layers with four convolutional layers, two max-pooling layers and two fully-connected layers. For all details, including training parameters, we refer the reader to (Carlini & Wagner, 2016a). On ImageNet we use the pretrained networks VGG-19 (Simonyan & Zisserman, 2014), ResNet-50 (He et al., 2015) and Inception-v3 (Szegedy et al., 2015) provided by Keras[5].

We evaluate the Boundary Attack in two settings: an (1) *untargeted setting* in which the adversarial perturbation flips the label of the original sample to any other label, and a (2) *targeted setting* in which the adversarial flips the label to a specific target class. In the untargeted setting we compare the Boundary Attack against three gradient-based attack algorithms:

- **Fast-Gradient Sign Method (FGSM).** FGSM is among the simplest and most widely used untargeted adversarial attack methods. In a nutshell, FGSM computes the gradient $g = \nabla_o \mathcal{L}(o, c)$ that maximizes the loss $\mathcal{L}$ for the true class-label $c$ and then seeks the smallest $\epsilon$ for which $o + \epsilon \cdot g$ is still adversarial. We use the implementation in Foolbox 0.10.0 (Rauber et al., 2017).

- **DeepFool.** DeepFool is a simple yet very effective attack. In each iteration it computes for each class $\ell \neq \ell_0$ the minimum distance $d(\ell, \ell_0)$ that it takes to reach the class boundary by approximating the model classifier with a linear classifier. It then makes a corresponding step in the direction of the class with the smallest distance. We use the implementation in Foolbox 0.10.0 (Rauber et al., 2017).

- **Carlini & Wagner.** The attack by Carlini & Wagner (Carlini & Wagner, 2016a) is essentially a refined iterative gradient attack that uses the Adam optimizer, multiple starting points, a tanh-nonlinearity to respect box-constraints and a max-based adversarial constraint function. We use the original implementation provided by the authors with all hyperparameters left at their default values[4].

To evaluate the success of each attack we use the following metric: let $\boldsymbol{\eta}_{A,M}(o_i) \in \mathbb{R}^N$ be the adversarial perturbation that the attack $A$ finds on model $M$ for the $i$-th sample $o_i$. The total score $\mathcal{S}_A$ for $A$ is the median squared L2-distance across all samples,

$$\mathcal{S}_A(M) = \underset{i}{\text{median}} \left( \frac{1}{N} \|\boldsymbol{\eta}_{A,M}(o_i)\|_2^2 \right). \tag{4}$$

For MNIST and CIFAR we evaluate 1000 randomly drawn samples from the validation set, for ImageNet we use 250 images.

### 3.1  UNTARGETED ATTACK

In the untargeted setting an adversarial is any image for which the predicted label is different from the label of the original image. We show adversarial samples synthesized by the Boundary Attack for each dataset in Figure 3. The score (4) for each attack and each dataset is as follows:

|  | Attack Type | MNIST | CIFAR | ImageNet | | |
|---|---|---|---|---|---|---|
|  |  |  |  | VGG-19 | ResNet-50 | Inception-v3 |
| FGSM | gradient-based | 4.2e-02 | 2.5e-05 | 1.0e-06 | 1.0e-06 | 9.7e-07 |
| DeepFool | gradient-based | 4.3e-03 | 5.8e-06 | 1.9e-07 | 7.5e-08 | 5.2e-08 |
| Carlini & Wagner | gradient-based | 2.2e-03 | 7.5e-06 | 5.7e-07 | 2.2e-07 | 7.6e-08 |
| Boundary (ours) | decision-based | 3.6e-03 | 5.6e-06 | 2.9e-07 | 1.0e-07 | 6.5e-08 |

---

[4]https://github.com/carlini/nn_robust_attacks (commit 1193c79)
[5]https://github.com/fchollet/keras (commit 1b5d54)

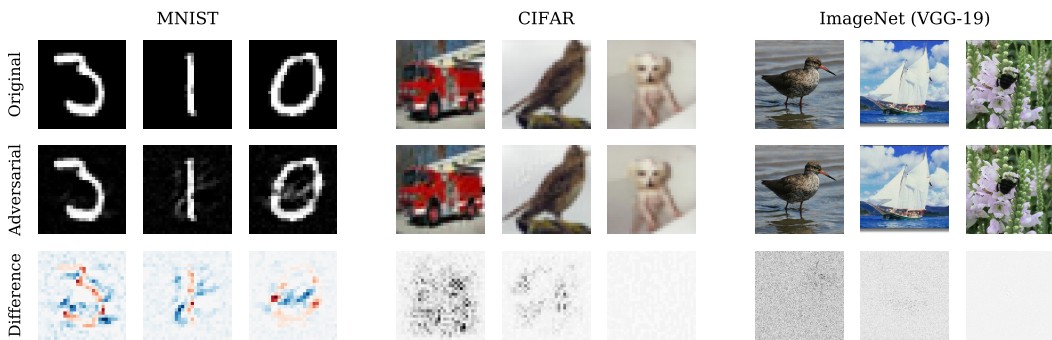

Figure 3: Adversarial examples generated by the Boundary Attack for an MNIST, CIFAR and ImageNet network. For MNIST, the difference shows positive (blue) and negative (red) changes. For CIFAR and ImageNet, we take the norm across color channels. All differences have been scaled up for improved visibility.

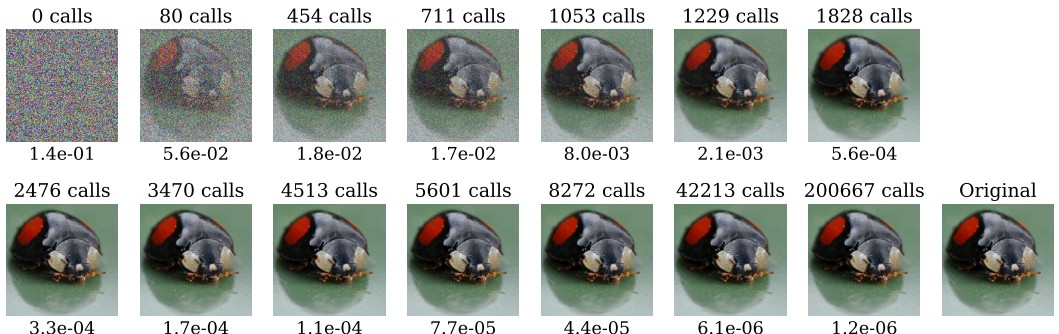

Figure 4: Example of an untargeted attack. Here the goal is to synthesize an image that is as close as possible (in L2-metric) to the original image while being misclassified (the original image is correctly classified). For each image we report the total number of model calls (predictions) until that point (above the image) and the mean squared error between the adversarial and the original (below the image).

Despite its simplicity the Boundary Attack is competitive with gradient-based attacks in terms of the minimal adversarial perturbations and very stable against the choice of the initial point (Figure 5). This finding is quite remarkable given that gradient-based attacks can fully observe the model whereas the Boundary Attack is severely restricted to the final class prediction. To compensate for this lack of information the Boundary Attack needs many more iterations to converge. As a rough measure for the run-time of an attack independent of the quality of its implementation we tracked the number of forward passes (predictions) and backward passes (gradients) through the network requested by each of the attacks to find an adversarial for ResNet-50: averaged over 20 samples and under the same conditions as before, DeepFool needs about 7 forward and 37 backward passes, the Carlini & Wagner attack requires 16.000 forward *and* the same number of backward passes, and the Boundary Attack uses 1.200.000 forward passes but zero backward passes. While that (unsurprisingly) makes the Boundary Attack more expensive to run it is important to note that the Boundary Attacks needs much fewer iterations if one is only interested in imperceptible perturbations, see figures 4 and 6.

## 3.2 TARGETED ATTACK

We can also apply the Boundary Attack in a targeted setting. In this case we initialize the attack from a sample of the target class that is correctly identified by the model. A sample trajectory from the starting point to the original sample is shown in Figure 7. After around $10^4$ calls to the model

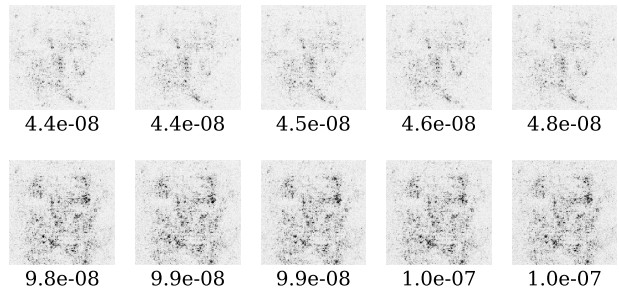

| 4.4e-08 | 4.4e-08 | 4.5e-08 | 4.6e-08 | 4.8e-08 |
| 9.8e-08 | 9.9e-08 | 9.9e-08 | 1.0e-07 | 1.0e-07 |

Figure 5: Adversarial perturbation (difference between the adversarial and the original image) for ten repetitions of the Boundary Attack on the **same image**. There are basically two different minima with similar distance (first row and second row) to which the Boundary Attack converges.

Figure 6: Distance between adversarial and original image over number of model calls for 12 **different images** (until convergence). Very few steps are already sufficient to get almost imperceptible perturbations.

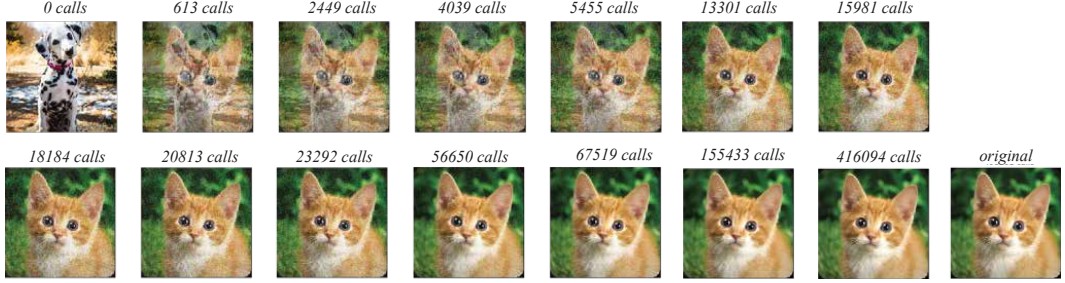

*0 calls* *613 calls* *2449 calls* *4039 calls* *5455 calls* *13301 calls* *15981 calls*

*18184 calls* *20813 calls* *23292 calls* *56650 calls* *67519 calls* *155433 calls* *416094 calls* *original*

Figure 7: Example of a targeted attack. Here the goal is to synthesize an image that is as close as possible (in L2-metric) to a given image of a tiger cat (2nd row, right) but is classified as a dalmatian dog. For each image we report the total number of model calls (predictions) until that point.

the perturbed image is already clearly identified as a cat by humans and contains no trace of the Dalmatian dog, as which the image is still classified by the model.

In order to compare the Boundary Attack to Carlini & Wagner we define the target target label for each sample in the following way: on MNIST and CIFAR a sample with label $\ell$ gets the target label $\ell + 1$ modulo 10. On ImageNet we draw the target label randomly but consistent across attacks. The results are as follows:

|  | Attack Type | MNIST | CIFAR | VGG-19 |
| --- | --- | --- | --- | --- |
| Carlini & Wagner | gradient-based | 4.8e-03 | 3.0e-05 | 5.7e-06 |
| Boundary (ours) | decision-based | 6.5e-03 | 3.3e-05 | 9.9e-06 |

## 4 THE IMPORTANCE OF DECISION-BASED ATTACKS TO EVALUATE MODEL ROBUSTNESS

As discussed in the introduction, many attack methods are straight-forward to defend against. One common nuisance is gradient masking in which a model is implicitly or explicitly modified to yield masked gradients. An interesting example is the saturated sigmoid network (Nayebi & Ganguli, 2017) in which an additional regularization term leads the sigmoid activations to saturate, which in turn leads to vanishing gradients and failing gradient-based attacks (Brendel & Bethge, 2017).

Another example is defensive distillation (Papernot et al., 2016). In a nutshell defensive distillation uses a temperature-augmented softmax of the type

$$softmax(x, T)_i = \frac{e^{x_i/T}}{\sum_j e^{x_j/T}} \tag{5}$$

and works as follows:

1. Train a teacher network as usual but with temperature $T$.
2. Train a distilled network—with the same architecture as the teacher—on the softmax outputs of the teacher. Both the distilled network and the teacher use temperature $T$.
3. Evaluate the distilled network at temperature $T = 1$ at test time.

Initial results were promising: the success rate of gradient-based attacks dropped from close to 100% down to 0.5%. It later became clear that the distilled networks only appeared to be robust because they masked their gradients of the cross-entropy loss (Carlini & Wagner, 2016b): as the temperature of the softmax is decreased at test time, the input to the softmax increases by a factor of $T$ and so the probabilities saturate at $0$ and $1$. This leads to vanishing gradients of the cross-entropy loss w.r.t. to the input on which gradient-based attacks rely. If the same attacks are instead applied to the logits the success rate recovers to almost $100\%$ (Carlini & Wagner, 2016a).

Decision-based attacks are immune to such defences. To demonstrate this we here apply the Boundary Attack to two distilled networks trained on MNIST and CIFAR. The architecture is the same as in section 3 and we use the implementation and training protocol by (Carlini & Wagner, 2016a) which is available at `https://github.com/carlini/nn_robust_attacks`. Most importantly, we do not operate on the logits but provide only the class label with maximum probability to the Boundary Attack. The results are as follows:

|  | Attack Type | MNIST | | CIFAR | |
| --- | --- | --- | --- | --- | --- |
|  |  | standard | distilled | standard | distilled |
| FGSM | gradient-based | 4.2e-02 | fails | 2.5e-05 | fails |
| Boundary (ours) | decision-based | 3.6e-03 | 4.2e-03 | 5.6e-06 | 1.3e-05 |

The size of the adversarial perturbations that the Boundary Attack finds is fairly similar for the distilled and the undistilled network. This demonstrates that defensive distillation does not significantly increase the robustness of network models and that the Boundary Attack is able to break defences based on gradient masking.

## 5 ATTACKS ON REAL-WORLD APPLICATIONS

In many real-world machine learning applications the attacker has no access to the architecture or the training data but can only observe the final decision. This is true for security systems (e.g. face identification), autonomous cars or speech recognition systems like Alexa or Cortana.

In this section we apply the Boundary Attack to two models of the cloud-based computer vision API by Clarifai[6]. The first model identifies brand names in natural images and recognizes over 500 brands. The second model identifies celebrities and can recognize over 10.000 individuals. Multiple identifications per image are possible but we only consider the one with the highest confidence score. It is important to note that Clarifai does provide confidence scores for each identified class (but not for all possible classes). However, in our experiments we do not provide this confidence score to the Boundary Attack. Instead, our attack only receives the name of the identified object (e.g. *Pepsi* or *Verizon* in the brand-name detection task).

We selected several samples of natural images with clearly visible brand names or portraits of celebrities. We then make a square crop and resize the image to $100 \times 100$ pixels. For each sample we make sure that the brand or the celebrity is clearly visible and that the corresponding Clarifai

---

[6]`www.clarifai.com`

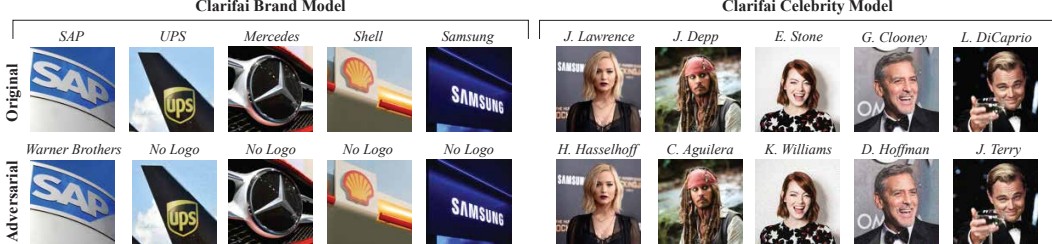

Figure 8: Adversarial examples generated by the Boundary Attack for two black-box models by Clarifai for brand-detection (left side) and celebrity detection (right side).

model correctly identifies the content. The adversarial criterion was misclassification, i.e. Clarifai should report a different brand / celebrity or None on the adversarially perturbed sample.

We show five samples for each model alongside the adversarial image generated by the Boundary Attack in Figure 8. We generally observed that the Clarifai models were more difficult to attack than ImageNet models like VGG-19: while for some samples we did succeed to find adversarial perturbations of the same order ($1e^{-7}$) as in section 3 (e.g. for *Shell* or *SAP*), most adversarial perturbations were on the order of $1e^{-2}$ to $1e^{-3}$ resulting in some slightly noticeable noise in some adversarial examples. Nonetheless, for most samples the original and the adversarial image are close to being perceptually indistinguishable.

## 6  DISCUSSION & OUTLOOK

In this paper we emphasised the importance of a mostly neglected category of adversarial attacks—*decision-based attacks*—that can find adversarial examples in models for which only the final decision can be observed. We argue that this category is important for three reasons: first, attacks in this class are highly relevant for many real-world deployed machine learning systems like autonomous cars for which the internal decision making process is unobservable. Second, attacks in this class do not rely on substitute models that are trained on similar data as the model to be attacked, thus making real-world applications much more straight-forward. Third, attacks in this class have the potential to be much more robust against common deceptions like gradient masking, intrinsic stochasticity or robust training.

We also introduced the first effective attack in this category that is applicable to general machine learning algorithms and complex natural datasets: the *Boundary Attack*. At its core the Boundary Attack follows the decision boundary between adversarial and non-adversarial samples using a very simple rejection sampling algorithm in conjunction with a simple proposal distribution and a dynamic step-size adjustment inspired by Trust Region methods. Its basic operating principle—starting from a large perturbation and successively reducing it—inverts the logic of essentially all previous adversarial attacks. Besides being surprisingly simple, the Boundary attack is also extremely flexible in terms of the possible adversarial criteria and performs on par with gradient-based attacks on standard computer vision tasks in terms of the size of minimal perturbations.

The mere fact that a simple constrained iid Gaussian distribution can serve as an effective proposal perturbation for each step of the Boundary attack is surprising and sheds light on the brittle information processing of current computer vision architectures. Nonetheless, there are many ways in which the Boundary attack can be made even more effective, in particular by learning a suitable proposal distribution for a given model or by conditioning the proposal distribution on the recent history of successful and unsuccessful proposals.

Decision-based attacks will be highly relevant to assess the robustness of machine learning models and to highlight the security risks of closed-source machine learning systems like autonomous cars. We hope that the Boundary attack will inspire future work in this area.

ACKNOWLEDGMENTS

This work was supported by the Carl Zeiss Foundation (0563-2.8/558/3), the Bosch Forschungsstiftung (Stifterverband, T113/30057/17), the International Max Planck Research School for Intelligent Systems (IMPRS-IS), the German Research Foundation (DFG, CRC 1233, Robust Vision: Inference Principles and Neural Mechanisms) and the Intelligence Advanced Research Projects Activity (IARPA) via Department of Interior/Interior Business Center (DoI/IBC) contract number D16PC00003. The U.S. Government is authorized to reproduce and distribute reprints for Governmental purposes notwithstanding any copyright annotation thereon. Disclaimer: The views and conclusions contained herein are those of the authors and should not be interpreted as necessarily representing the official policies or endorsements, either expressed or implied, of IARPA, DoI/IBC, or the U.S. Government.

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
