# OpenReview forum: "Decision-Based Adversarial Attacks: Reliable Attacks Against Black-Box Machine Learning Models"
_ICLR.cc/2018/Conference — Accept (Poster)_

### Official Review · AnonReviewer2 · 2017-11-27
**Innovative new class of attacks on neural networks, which pose a threat to distillation defenses**

**Rating:** 7
**Confidence:** 4

**Review:**

The authors identify a new security threat for deep learning: Decision-based adversarial attacks. This new class of attacks on deep learning systems requires from an attacker only the knowledge of class labels (previous attacks required more information, e.g., access to a gradient oracle). Unsurprisingly, since the attacker has so few information, such kind of attacks involves quite a lot trial and error. The authors propose one specific attack instance out of this class of attacks. It works as follows.

First, an initial point outside of the benign region is guessed. Then multiple steps towards the decision boundary is taken, finally reaching the boundary (I am not sure about the precise implementation, but it seems not crucial; the author may please check whether their description of the algorithm is really reproducable). Then, in a nutshell, a random walk on a sphere centered around the original, benign point is performed, where after each step, the radius of the sphere is slightly reduced (drawing the point closer to the original point), if and only if the resulting point still is outside of the benign region.

The algorithm is evaluated on the following datasets: MNIST, CIFAR, VGG19, ResNet50, and InceptionV3.

The paper is rather well written and structured. The text was easy to follow. I suggest that a self-contained description of the problem setting (assumptions on attacker and defender; aim?) shall be added to the camera-ready version (being not familiar with the area, I had to read a couple of papers to get a feeling for the setting, before reviewing this paper). As in many DL papers these days, there really isn't any math in it worth a mention; so no reason here to say anything about mathematical soundness. The authors employ a reasonable evaluation criterion in their experiments: the median squared Euclidean distance between the original and adversarially modified data point. The results show consistent improvement for most data sets.

In summary, this is an innovative paper, proposing a new class of attacks that totally makes sense in my opinion. Apart from some minor weaknesses in the presentation that can be easily fixed for the camera ready, this is a nice, fresh paper, that might spur more attacks (and of course new defenses) from the new class of decision-based attacks. It is worth to note that the authors show that distillation is not a useful defense against such attacks, so we may expect follow-up proposing useful defenses against the new attack (which BTW is shown to be about a factor of 10 in terms of iterations more costly than the SOTA).

---

> ### Author Response · Authors · 2017-12-21
> **Response to the second reviewer**
>
> Thanks a lot for your positive review!
>
> We appreciate your nice summary of the attack. To find the first point „on“ (i.e. close to) the boundary, we just perform a line search between the initial point and the original. As you said, this is indeed not crucial and it would be perfectly fine to leave out this step and directly follow the actual algorithm of the Boundary Attack. We use the line search because it basically just gives us a better starting point and therefore speeds up the beginning of the attack.
>
> Regarding your first sentence, we’d like to add that the Boundary Attack does not just pose a new threat, it also makes measuring the robustness of models more reliable. In the past virtually all defence strategies proposed in the literature did not actually increase the robustness of the model but only disabled the attack strategy. A prime example is gradient masking where the backpropagated gradient needed in gradient-based attacks is simply zero (e.g. defensive distillation). The Boundary Attack is robust to many such nuisances and will make it easier to spot whether a model is truly robust.
>
> We improved the introduction to make the paper more self-contained and thus easier to to read for people not familiar with the area. We’d appreciate your feedback if something is still unclear.

---

### Official Review · AnonReviewer3 · 2017-11-27
**Old problem formulation, but interesting empirical results in a new application domain**

**Rating:** 7
**Confidence:** 4

**Review:**

This is a nice paper proposing a simple but effective heuristic for generating adversarial examples from class labels with no gradient information or class probabilities. Highly relevant prior work was overlooked and there is no theoretical analysis, but I think this paper still makes a valuable contribution worth sharing with a broader audience.

What this paper does well:
- Suggests a type of attack that hasn't been applied to image classifiers
- Proposes a simple heuristic method for performing this attack
- Evaluates the attack on both benchmark neural networks and a commercial system

Problems and limitations:

1. No theoretical analysis. Under what conditions does the boundary attack succeed or fail? What geometry of the classification boundaries is necessary? How likely are those conditions to hold? Can we measure how well they hold on particular networks?

Since there is no theoretically analysis, the evidence for effectiveness is entirely empirical. That weakens the paper and suggests an important area of future work, but I think the empirical evidence is sufficient to show that there's something interesting going. Not a fatal flaw.

2. Poor framing. The paper frames the problem in terms of "machine learning models" in general (beginning with the first line of the abstract), but it only investigates image classification. There's no particular reason to believe that all machine learning algorithms will behave like convolutional neural network image classifiers. Thus, there's an implicit claim of generality that is not supported.

This is a presentation issue that is easily fixed. I suggest changing the title to reflect this, or at least revising the abstract and introduction to make the scope clearer.

A minor presentation quibble/suggestion: "adversarial" is used in this paper to refer to any class that differs from the true class of the instance to be disguised. But an image of a dalmation that's labeled as a dalmation isn't adversarial -- it's just a different image that's labeled correctly. The adversarial process is about constructing something that will be mislabeled, exploiting some kind of weakness that doesn't show up on a natural distribution of inputs. I suggest rewording some of the mentions of adversarial.

3. Ignorance of prior work. Finding deceptive inputs using only the classifier output has been done by Lowd and Meek (KDD 2005) for linear classifiers and Nelson et al. (AISTATS 2010, JMLR 2012) for convex-inducing classifiers. Both works include theoretical bounds on the number of queries required for near-optimal adversarial examples. Biggio et al. (ECML 2013) further propose training a surrogate classifier on similar training data, using the predictions of the target classifier to relabel the training data. In this way, decision information from the target model is used to help train a more similar surrogate, and then attacks can be transferred from the surrogate to the target.

Thus, "decision-based attacks" are not new, although the algorithm and experiments in this paper are.


Overall, I think this paper makes a worthwhile contribution, but needs to revise the claims to match what's done in the paper and what's been done before.

---

> ### Author Response · Authors · 2017-12-21
> **Response to the third reviewer**
>
> Thanks a lot for your insightful comments!
>
> 1. We agree that a theoretical analysis of the decision boundaries of neural networks is an important area of future work. One assumption of the attack is that the boundary is fairly smooth, otherwise the linearity assumption in the proximity of the current adversarial wouldn’t hold. Other then that the success of the attack very much depends on the number and distribution of local minimas on the boundary, but that’s true for all attacks. So just as the success of stochastic gradient descent is first and foremost an empirical result, so is the success of the Boundary Attack and it will be challenging (yet interesting) to get better theoretical insights in the future.
>
> 2. The Boundary Attack is by no means restricted to CNNs. However, virtually all papers on adversarial attacks from recent years were evaluated on CNNs in computer vision tasks and so we followed this setup in order to be as accountable as possible. Of course, we do not claim that the Boundary Attack will work equally well on all Machine Learning algorithms, but that’s something no attack can legitimately claim (including gradient-based attacks). In other words, the Boundary Attack is in principle applicable to any machine learning algorithm but how well it will perform is an empirical question that future work will highlight. We’d thus choose to leave the title as is in order to stimulate more progress in this direction.
>
> 3. We thank you for these pointers to relevant prior work which have prompted us to adapt the claims of the manuscript accordingly. More concretely, we now delineate our work more clearly as the first decision-based attack that scales to complex machine learning algorithms (such as DNNs) and complex data sets (such as ImageNet).
>
> We also added more explanations as to our definition of an “adversarial”.

---

### Official Review · AnonReviewer1 · 2017-11-30
**Well written paper with interesting results.**

**Rating:** 8
**Confidence:** 3

**Review:**

In this paper, the authors propose a novel method for generating adversarial examples when the model is a black-box and we only have access to its decisions (and a positive example).  It iteratively takes steps along the decision boundary while trying to minimize the distance to the original positive example.


Pros:
- Novel method that works under much stricter and more realistic assumptions.
- Fairly thorough evaluation.
- The paper is clearly written.


Cons:
- Need a fair number of calls to generate a small perturbation.  Would like to see more analysis of this.
- Attack works for making something outside the boundary (not X), but is less clear how to generate image to meet a specific classification (X).  3.2 attempts this slightly by using an image in the class, but is less clear for something like FaceID.
- Unclear how often the images generated look reasonable.  Do different random initializations given different quality examples?

---

> ### Author Response · Authors · 2017-12-21
> **Response to the first reviewer**
>
> Thanks a lot for the positive review! We have added two new figures to address your comments (figures 5 and 6).
>
> - It is true that the Boundary Attack needs a fair number of calls until it converges to the absolute minimal perturbation. In most practical scenarios, however, it is sufficient to generate adversarials that are perceptually indistinguishable from the original image in which case a much smaller number of iterations is necessary. We added figure 6 to show how fast the size of the perturbation is reduced by the Boundary Attack.
>
> - The Boundary Attack is made to find minimal perturbations from a given image that is misclassified (or classified as a certain target). More graphically, it tries to find an image that is classified as a bus but clearly looks like a cat. In the case you mentioned - Face ID - one would try to find any image that is classified as a certain target. In other words: you’d try to find an image that is classified as a bus, and it’s completely fine if it also looks like a bus to humans. That’s a totally different threat scenario that we do not consider in the paper.
>
> - All results shown in the paper are for a single run of the Boundary Attack, there is no cherry picking. To make this point clearer we added figure 5 where we show the result of ten runs of the Boundary Attack on the same image with random initialization. The final perturbation ends up in one of two minima, and both are of similar quality (the size of the perturbations varies only by a factor of 2).

---

### Public Comment · ~Claude_Chen1 · 2017-11-11
**Implementation Request**

Hi:
We are students at Carnegie Mellon University participating the ICLR 2018 Reproducibility Challenge. We find this paper quite interesting and were wondering if it is possible for you to share the implementation of boundary attack.

Much appreciated!

---

> ### Author Response · Authors · 2017-11-24
> **Implementation will be released soon**
>
> Hi,
>
> the Reproducibility Challenge is a great idea!
> We will make our implementation available on GitHub as soon as we have cleaned it up in a way that makes it easy to apply.
>
> We will also post a link here and update the paper accordingly once the double-blind review period has ended.

---

> > ### Author Response · Authors · 2017-12-08
> > **Source Code Released**
> >
> > Our implementation of the Boundary Attack is now available at http://bit.ly/2kF0JKQ.
> > We post a short URL because of the double-blind review.

---

### Public Comment · (anonymous) · 2017-12-10
**A few questions regarding the submission's threat model**

I enjoyed reading your submission and found that the attack proposed is interesting. I however found that the presentation and scope of the paper claims misrepresent results included in the evaluation.

The draft does not include a threat model describing what you consider to be an adversarial example.  This is especially important given that the only metric used in the evaluation, which you proposed in Equation 4, differs from metrics used by all previous work in the literature. Given that you only report the median, could you comment on the maximum L2 perturbation norm of adversarial inputs your approach produces in your experiments? How do you guarantee that all inputs produced are individually adversarial? In the literature, attack papers report the success rate in addition to the perturbation norm, which helps better evaluate their effectiveness.

Could you elaborate on the cost of your attacks? You make the strong claim that the attack is easier to mount than a transfer-based attack and "performs on par with gradient-based attacks" as written in the conclusion. Yet, producing a single adversarial example with the attack proposed requires more queries than needed to train a substitute model in a transfer-based attack or compute the gradients in a gradient-based attack. This means that if the adversary is interested in scaling the attack to find multiple inputs misclassified by the model, it will potentially need to make millions of additional queries. Furthermore, your algorithm is initialized with an input that is already adversarial. Do you include the queries needed to find such an input in the cost of your attack?

Could you also comment on the number of queries needed to evade the Clarifai model? Where these attacks targeted or untargeted? How many adversarial examples did you produce that successfully attacked the Clarifai model?

Related to the attack's cost, have you considered how the attack will fare against detection by the defender: all queries made are very similar to each other so it seems like a natural defense strategy would be to reject the queries based on their large number and similarity to one-another.

I also found several inconsistencies, which you might be able to clarify:

In the abstract, you state that transfer-based attacks "need access to the training data" but later point to a work that does not need this adversarial capability [http://doi.acm.org/10.1145/3052973.3053009].

In the introduction, you state that gradient-based attacks can be defended against by masking gradients. However, this is wrong as pointed out by prior work that you cited [http://arxiv.org/abs/1607.04311, http://doi.acm.org/10.1145/3052973.3053009, http://arxiv.org/abs/1704.01547].

When comparing to prior work in decision-based attacks [http://doi.acm.org/10.1145/3052973.3053009], you state that "While this approach works well on MNIST it has yet to be shown that it scales to more complex natural datasets such as CIFAR or ImageNet." However, the related work you cite also includes experiments on a dataset of traffic sign images (GTRSRB).

The draft claims that the attack works on all machine learning models but only evaluates on CNNs. It might make sense to reconsider the title.

---

> ### Author Response · Authors · 2017-12-19
> **Clarifications**
>
> > The draft does not include a threat model describing what you consider to be an adversarial example.
>
> As stated in the manuscript we consider an adversarial example to be any image that is differently classified as the original image. Most importantly, this definition is independent of human perception but is only relative to the original image. To goal of the attack is to make the difference between the adversarial and the original image as small as possible.
>
> > Given that you only report the median, could you comment on the maximum L2 perturbation norm of adversarial inputs your approach produces in your experiments?
>
> For VGG-19 on ImageNet in the untargeted setting the maximum L2 perturbation is as follows:
>
> FGSM: 2.2e-2
> DeepFool: 1.8e-4
> Boundary Attack: 7.1e-5
> Carlini & Wagner: 4.8e-5
>
> > How do you guarantee that all inputs produced are individually adversarial?
>
> The Boundary Attack starts from an adversarial image and stays within the adversarial region. Thus, the result of the attack is guaranteed to be adversarial.
>
> > In the literature, attack papers report the success rate in addition to the perturbation norm, which helps better evaluate their effectiveness.
>
> I find the success rate to be fairly meaningless: an attack should always be successful if it works correctly, the question is just how large the perturbation needs to be until one gets an adversarial.
>
> >  producing a single adversarial [...] requires more queries than needed to train a substitute model [...] or compute the gradients in a gradient-based attack.
>
> Obviously gradient-based attacks need fewer model calls, but that’s simply because the gradient yields a lot of information about the model. Training a substitute model doesn’t need any calls to the original model but then there are no guarantees that the adversarials will transfer (they do empirically, but that might change with future architectures). Furthermore, a few thousand calls are totally sufficient for the Boundary Attack to produce adversarials that can hardly be distinguished from the original image. One just needs millions of queries if one really tries to find the absolute minimum perturbation.
>
> > Furthermore, your algorithm is initialized with an input that is already adversarial.
>
> Finding an initial adversarial is easy: just take any image from a different class. Of course, the distance to the original image is high initially, but that changes over the course of the optimisation. I think the confusion here is related to how we define adversarial images (see above). We’ll clarify that more clearly in the manuscript.
>
> > Could you comment on the number of queries needed to evade the Clarifai model?
>
> For each image around 2000 - 4000 calls where needed. The attack was untargeted and we only produced the images you find in the article.
>
> > a natural defense strategy would be to reject the queries based on their large number and similarity to one-another.
>
> Surely with proper engineering one can have a chance to detect this attack (but also to evade the defence by properly distributing the calls over clients and time), but this argument can be used against basically all attacks other then transfer attacks (which have their own problems) and FGSM (if one has access to gradients). On a different note, the Boundary Attack is not only about security but also about evaluating the robustness of models as it is not so easily fooled by things like gradient masking.
>
> > In the abstract, you state that transfer-based attacks "need access to the training data" but later point to a work that does not need this adversarial capability [http://doi.acm.org/10.1145/3052973.3053009].
>
> The work you cite is not a classical transfer attack but we qualify it as a decision-based attack because it needs strictly less knowledge then transfer attacks.
>
> > In the introduction, you state that gradient-based attacks can be defended against by masking gradients. However, this is wrong as pointed out by prior work that you cited.
>
> That prior work makes precisely the point that gradient-based attacks fail due to gradient masking (that’s basically by definition!). The papers show this by making certain interventions that remove gradient masking (after which the gradient-based attacks work again).
>
> > However, the related work you cite also includes experiments on a dataset of traffic sign images (GTRSRB).
>
> We’ve extended this sentence to include the traffic sign images. Basically, the variant of transfer attacks you are pointing to will work on any dataset for which the intra-class variability is low, but this is not true for CIFAR and ImageNet (and most other interesting datasets).
>
> > The draft claims that the attack works on all machine learning models but only evaluates on CNNs.
>
> There is nothing in the Boundary Attack that restricts it to CNNs. CNNs are just the class of models that basically all relevant prior work has been evaluated on, and so we follow this direction.

---

### Author Response · Authors · 2017-12-22
**Summary of the updates to the manuscript**

We have uploaded an updated version of the paper with the following changes:

1.) The Boundary attack has now been run to full convergence for all experiments reported in the paper. On some experiments, in particular on ImageNet, the performance considerably increased and now places the attack squarely between DeepFool and Carline & Wagner in all setups.

2.) We added a new figure showing the result of the Boundary Attack for repeated runs on the same sample. The attack converges to only two minima with similar distance to the original image (Figure 5).

3.) We added a new figure showing the distance to the original image as a function of model calls (Figure 6). The figure further highlights that relatively few model calls are already sufficient to find excellent adversarial examples. A large number of calls is only necessary to fully converge.

4.) We improved the introduction to be more self-contained and easier to read and added additional references to prior work.

---

### Public Comment · (anonymous) · 2018-01-09
**prior work**

Very interesting paper!
I found the idea quite similar with the paper:https://ix.cs.uoregon.edu/~lowd/kdd05lowd.pdf
Both modify the malicious instance towards benign heuristically, except that the prior work have theoretic guarantees for simple models and binary features.
Given this, I think the novelty of the paper is reduced. Can the authors help to make further comparisons?
Thanks a lot!

---

> ### Author Response · Authors · 2018-01-10
> **Prior work limited to linear classifiers**
>
> Thanks for your comment. The attack you mention is strictly limited to linear classifiers whereas our method is applicable to highly nonlinear classifiers like deep neural networks. That generality makes theoretical guarantees almost impossible. A close analogy is gradient descent: optimizing linear classifiers (convex) comes with convergence guarantees whereas optimizing neural networks (non-convex) comes with no guarantees at all.

---

### Decision · Program_Chairs · 2018-01-29
**ICLR 2018 Conference Acceptance Decision**

**Decision:**

Accept (Poster)

**Comment:**

The reviewers all agree this is a well written and interesting paper describing a novel black box adversarial attack.   There were missing relevant references in the original submission, but these have been added.  I would suggest the authors follow the reviewer suggestions on claims of generality beyond CNN; although there may not be anything obvious stopping this method from working more generally, it hasn't been tested in this work.   Even if you keep the title you might be more careful to frame the body in the context of CNN's.